# From Seaweeds to Hydrogels: Recent Progress in Kappa-2 Carrageenans

**DOI:** 10.3390/ma16155387

**Published:** 2023-07-31

**Authors:** Hiléia K. S. Souza, Wala Kraiem, Amine Ben Yahia, Adel Aschi, Loïc Hilliou

**Affiliations:** 1Institute for Polymers and Composites (IPC), Campus de Azurém, University of Minho, 5800-048 Guimarães, Portugal; hileia.souza@dep.uminho.pt (H.K.S.S.); wala.kraiem@fst.utm.tn (W.K.); amine.benyahia@fst.utm.tn (A.B.Y.); 2Centre for Innovation in Polymer Engineering (PIEP), Campus de Azurém, University of Minho, Building 15, 4800-058 Guimarães, Portugal; 3LR99ES16 Laboratoire Physique de la Matière Molle et de la Modélisation Électromagnétique, Faculté des Sciences de Tunis, Université de Tunis El Manar, Tunis 2092, Tunisia; aschi13@yahoo.fr

**Keywords:** hybrid carrageenan, kappa-2 carrageenan, polysaccharide, seaweed, hydrogel, rheology

## Abstract

Hybrid carrageenans, also called kappa-2 (K2) or weak kappa, are a class of sulfated polysaccharides with thermo-reversible gelling properties in water and are extracted from a specific family of red seaweeds. K2 are known in the industry for their texturizing properties which are intermediate between those of kappa-carrageenans (K) and iota-carrageenans (I). As such, K2 are gaining industrial interest, as they can replace blends of K and I (K + I) in some niche applications. Over the last decade or so, some progress has been made in unravelling K2′s chemical structure. The understanding of K2 gel’s structure–rheological properties’ relationships has also improved. Such recent progress is reported here, reviewing the literature on gelling K2 published since the last review on the topic. The focus is on the seaweeds used for extracting K2, their block copolymer chemical structures, and how these impact on the gel’s formation and rheological properties. The outcome of this review is that additional rheological and structural studies of K2 hydrogels are needed, in particular to understand their viscoelastic behavior under large deformation and to unravel the differences between the texturizing properties of K2 and K + I.

## 1. Introduction

Carrageenans are a family of sulfated polysaccharides extracted from Gigartinales, a specific order of red algae also labeled carrageenophytes. These natural biopolymers are used essentially as food additives (E407) for texturizing formulations [1,2]. More recently, carrageenans entered the cosmetics and pharmaceutics markets as excipient for capsules and tablets [2] and also for cell immobilization in drug production [3]. They also have good prospects for application in the nutraceutical and pharmaceutical industries [4] in spite of the controversy over their food safety [5,6]. Carrageenans’ gelling properties in water-based formulations enable the encapsulation of drugs for better delivery, and their bioactive attributes [7], in particular as antivirals against SARS-CoV-2 [8], are very attractive.

The food ingredients market usually classifies carrageenans into three categories based on their functionality: kappa-carrageenan (K), which forms strong gels, iota-carrageenan (I), which forms weaker gels; and lambda-carrageenan (L), which produces viscous solutions. A fourth class, hybrid carrageenans, also called kappa-2 (K2) or weak kappa [9], has gained scientific and industrial attention over the last 20 years [9,10,11,12]. The ideal chemical structures of the repeating disaccharide units found in the gelling carrageenans (i.e., K, I, and K2) are provided in Figure 1, where the nomenclature introduced by Knutsen et al. [13] is used. There are two main reasons for the increasing interest in K2. First, recent issues in farms of seaweeds cultivated for K and I production [14,15] have led to a significant drop in the production of carrageenans. As K2 are isolated from alternative carrageenophytes, they critically contribute to addressing the greater demand in the food sector [2]. Second, K2 are used in niche dairy applications to substitute mixes of intermediate strength K and I gels [9,16] but also to replace non-gelling carrageenan blends. Thus, the direct use of K2 instead of carrageenan blends avoids an additional costly mixing pro cess after carrageenan extraction [17].

Since the last reviews on K2 [10,11], some progress has been made in the understanding of their biosynthesis and chemical structure [18] and on how to tune it through specific extraction routes or seaweeds treatments. The understanding of K2s gel’s structure–rheological properties’ relationships has also improved. Such recent progress is reported here, reviewing the literature on gelling K2 published over the last decade or so. 

## 2. From Seaweeds to Hybrid Carrageenans

### 2.1. Seaweeds Used for Kappa-2 Carrageenan Production: The Carrageenophytes

Carrageenans are components of the seaweeds’ cell walls that primarily provide resistance to oceanic physical forces, osmotic stress (variation in salinity), and to desiccation during low tides, among other attributes (see, e.g., [19] and references therein). The current model for the biosynthetic pathway of carrageenan in seaweeds is described in three steps involving three different types of enzymes [18,19]: galactose residues are possibly polymerized first and then sulfated in the cells before the final cyclization of galactose 6-sulfate moieties into 3,6-anhydro derivatives occurring in the cell walls [20]. The enzymatic cyclization is now firmly established [18,19,20] and is thus illustrated in Figure 1. The main carrageenophytes used to produce K2 are listed in Table 1, taking into account the definition given to K2 in terms of chemical composition: carrageenans containing between 20 and 50 mol% of ***G4S-DA2S*** [9,12,16]. However, the list in Table 1 is extended to K2 containing between 80 and 50 mol% of ***G4S-DA***, based on the K2 chemical structures reviewed by van de Velde [10] and documented in more recent studies, and assuming that the remaining carrageenan fraction is essentially I. Also, a 5% error in the quantification of K is considered here, since it is the uncertainty of proton NMR spectroscopy often used to characterize the extracted K2.

In a seminal paper published three decades ago, Chopin et al. [21] performed the Fourier transform infrared diffuse reflectance spectroscopy (DRIFT) of powdered algal materials to achieve a direct qualitative assessment of the main K2 contained in carrageenophytes. The DRIFT spectra of seaweeds present symptomatic absorption bands at specific wavelengths which can be assigned to specific chemical groups of the carrageenan disaccharide units (see the wavelength numbers in Figure 1). DRIFT was then used with many carrageenanophytes [22] and other infrared spectroscopic techniques (including Raman) were used over the last decade for a fast assessment of seaweeds carrageenan constituents [23,24,25].

**Table 1 materials-16-05387-t001:** Gigartinales with the carrageenan constituents and corresponding chemical composition of K2 extracted using the listed protocol (alkali or water extraction). Data are taken from the references listed in the last column (Ref.), where blends of generic phases or reproductive plants are studied. The K content is given in mol.% computed from the NMR data reported in the corresponding reference study. L stands for lambda-carrageenan (***G2S-D2S,6S***) and T for theta-carrageenan (***G2S-DA2S***).

Seaweed Species	Carrageenans in Seaweeds	Extraction	K Content in K2	Ref.
*Ahnfeltiopsis devoniensis*	K, I, M	water	50–55	[26]
		alkali	17–35 ^a,b^	[27]
	K, I, M, N	alkali	30–50 ^c^	[28]
	K, I, M, N	water	42–50 ^c^	[28]
*Chondracanthus acicularis*	K, I	alkali	76	[10,22]
	K, I, M, N, L, T			[23]
	T			[24]
		water/alkali	60 ^a^	[27]
*Chondracanthus canaliculatus*	K, I ^c^	alkali	78	[10,22]
*Chondracanthus chamissoi*	K, I, T	alkali	82	[10,22]
		water	35–100 ^b^	[29]
		water	59	[30]
		alkali	64	[30]
*Chondracanthus corymbiferus*	K, I	alkali	74	[10,22]
*Chondracanthus teedei*	K, I, M, N, T	alkali	50	[24,25]
		water/alkali	53–58 ^a^	[27]
	K, I, N	alkali	50–58 ^c^	[10,22]
*Chondrus canaliculata*	K, L, T	alkali	69	[10,22]
*Chondrus crispus*	K, I, M, L, T ^c^	alkali	30–81 ^c^	[10,22]
	K, I, M, L			[24]
	K, I, M	water	60–70	[26]
		alkali	70 ^a^	[27]
		alkali	64 ^a^	[31]
		alkali	71–72	[32]
	K, M, N	alkali	65–84 ^c^	[28]
	K, M, N	water	65–75 ^c^	[28]
*Chondrus ocellatus*	K, I, L		85	[10,22]
*Eucheuma isiforme*	I	alkali	60	[10,22]
*Eucheuma platycladum*	K, I	alkali	83	[10,22]
*Gigartina alveata*	K, L	alkali	64	[10,22]
*Gigartina bracteata*	K, L			[10]
*Gigartina chamissoi*		alkali	54	[22]
		alkali	57	[32]
*Gigartina clathrata*		alkali	80	[22]
*Gigartina pistillata*	K, I, M, N, L			[24]
	K, I		41–45	[10,22]
	KIM, L, T			[23]
		alkali	35–49 ^a^	[27]
*Gigartina radula*	K, L	alkali	50	[10,22]
*Gigartina skottsbergii* ^d^	K, I, M, N, L ^c^	alkali	59	[10,22]
		alkali	57	[32]
		water	62	[33]
		water	70	[34]
*Gymnogongrus crenulatus*	K, I, L	alkali	64 ^c^	[10,22]
		alkali	60–64	[27]
*Gymnogongrus humilis*	K, I	alkali	68	[10,22]
*Gymnogongrus tenuis*		water	14–51 ^b,c^	[35]
*Gymnogongrus torulosus*		alkali	45	[10]
*Gymnogongrus vermicularis*		alkali	71	[10]
*Iridaea undulosa*		alkali	58–62	[10]
*Iridaea cordata*		water	42 ^a,b,c^–66 ^a^	[36]
*Mastocarpus pacificus*		water	>50	[37]
*Mastocarpus stellatus*	K, I, M, N, L ^c^	alkali	74	[10,22]
	K, I, M, N			[23,24]
	K, I, M, N	water	53–67	[26]
		alkali	62 ^a^	[27]
		water	55–64 ^a^	[38]
	K, M, N	alkali	62–80 ^a^	[39]
	K, M, N	water	62–65 ^a^	[39]
*Mazzaella laminarioides*	I, L	alkali	55–75 ^c^	[10,22]
	K, I, M, N			[40]
		water	46 ^b^	[30]
		alkali	54	[30]
*Mazzaella parksii*		water	63–79 ^c^	[41]
*Mazzaella splendens*	K, I, L, T	alkali	72	[10,22]
*Sarcothalia crispata*	K, I, M, N			[24]
	K, I, M, N, L, T ^c^	alkali	40–63	[10,22]
		water	51	[30]
		alkali	56	[30]
		water	56	[34]
*Sarcothalia radula*		water	45	[30]
		alkali	53	[30]
*Turnerella mertensiana*		water	65 ^c^	[42]
		alkali	82	[42]

^a^ Content from reproductive seaweeds. ^b^ Does not qualify for K2, based on the definition of K2 reported in [9,12,16]. ^c^ Agarocolloids and/or starch also reported. ^d^ Gigartina Skottsbergii is now Sarcopeltis Skottsbergii.

Table 1 compiles the carrageenan composition of some carrageenophytes, starting from the data reported by Chopin et al. [22] and focusing on more recent updates. Solid-state ^13^C NMR spectra measured on powdered algal materials has also been used for a qualitative assessment of carrageenans present in seaweeds [43,44]. However, as noted recently, such a technique is still underutilized in spite of showing good prospects for a quantitative assessment of the carrageenan composition of seaweeds [26]. The geographical variations that impact on the growing conditions of seaweeds lead to variations in the types of biosynthesized carrageenans among seaweeds populations, see for example refs. [27,31,38], as well as in the carrageenan content in seaweeds [45]. In addition, the reproductive and vegetative life stages of seaweeds, as listed in Table 1, are known to contain different types of carrageenans (see, e.g., [18,19,20,21,22,24,25,26,27,31,36,38,40,45,46] and references therein). Thus, several entries in Table 1 mirror such variation in the carrageenan composition of seaweeds, as well as in the K content of the extracted carrageenans. But this has virtually no impact on the K2 chemical structures produced in the industry from a specific carrageenophyte. This is because the large scale extraction process does not include a sorting of seaweeds, and alkali are often used to convert more sulfated disaccharide units into gelling K2 (see Figure 1).

In contrast to the direct chemical analysis of seaweeds, most of the research on the identification of the gelling K2 produced by carrageenophytes, which is reviewed elsewhere [10], relies on the extraction of polysaccharides from seaweeds. K2 extraction, thus, adds complexity to the classification of carrageenophytes for K2 production, as variability in the employed extraction protocols is intrinsically linked to the variability of the isolated K2 chemical structures. Thus, variability in the K content of K2 extracted from the seaweeds listed in Table 1 comes as no surprise to the extent that some reports suggest that certain extracted carrageenans do not qualify for the industrial definition of K2. The extraction routes for isolating K2 from carrageenophytes is detailed in the next section. 

### 2.2. Extraction of Hybrid Carrageenans

The extraction routes used in the studies referenced in Table 1 are shown in Figure 2. 

First, processes such as the drying and grinding of harvested seaweeds are ubiquitous in the industry before starting the extraction step. Few reports suggested that these postharvest processes affect the hybrid carrageenan characteristics. Higher temperatures used during the air drying of *M. stellatus* gametophytes lead to the recovery of K2 with untouched chemical structures but with smaller molecular masses [47]. The grinding of these seaweeds into smaller particles enabled the water extraction of more K2 (extraction yield increased) with larger molecular masses [48]. In contrast to this, for *C. crispus* gametophytes, the particle size did not impact on the extraction yield of the hybrid carrageenans [49]. Clearly, this topic deserves more research efforts. However, few studies avoid seaweeds drying and grinding to bypass any possible effect on extracted K2 characteristics, whereas larger K2 recovery yields are not needed [30,31,38]. Another process, the dark treatment of carrageenophytes during cultivation after harvesting, has also attracted little research, in contrast to the dark treatment of seaweeds used in the production of agar. The cultivation in the dark of *C. crispus* was shown to increase the amount of 3,6-anhydrogalactose in recovered K2 with correspondingly less sulfate content [50]. As such this postharvest process could be optimized for an ecofriendly in vivo enzymatic alternative to the alkali conversion of M and N into K and I [50,51]. Another ecofriendly alternative consisting in the postharvest culture of commercial seaweeds in low-nutrient condition has been suggested for the production of I [52], but this approach has not yet been tested for K2 carrageenophytes.

After seaweed postharvest treatment, K2 extraction is performed by dispersing algal material in water. Water extraction stems from the leaching of very different carrageenans (including non-gelling) from the seaweeds. Hybrid carrageenan solubility in water was recently shown to be strongly dependent on the type of cations present in the postharvested seaweeds, on the contacting duration *t* with water, and on the temperature, *T*, used during extraction [53]. Parameters *t* and *T*, as well as the type and amount of alkali, can thus be independently varied to tailor the chemical composition of the extracted K2. Lower *T* were consistently reported to yield more sulfated K2 for *C. chamissoi* [29] and *C. crispus* [53]. However, for *G. tenui*, a K2 was extracted with cold water, whereas a more sulfated hybrid carrageenan was recovered at 80 °C, which as such does not qualify as a K2 [35]. In presence of alkali, longer *t* was associated with the recovery of K2 containing less biological precursors (M and N) for *M. stellatus* [39], *C. crispus*, and *A. devoniensis* [28]. The latter two studies also showed that NaOH was more efficient than KOH to convert M and N into K and I disaccharide units, as shorter *t* were needed to achieve the minimum levels of more sulfated disaccharide units. Finally, the extraction parameters are also known to impact on the molecular masses of K2 [54,55], in particular with stronger and longer alkali treatments giving K2 a reduced chain size [28,39]. Overall, as both the kinetics and extent of the cyclization reactions are known to depend on K2′s chemistry [56], more studies on the optimization of the alkali extraction parameters towards the recovery of specific K2 chemical structures should be extended to other carrageenophytes.

After the extraction, solid algal residues are separated from the K2-rich solution, by filtration or centrifugation. Then, K2 is recovered from the solution usually by precipitation in alcohol, as is the case for all of the extracts listed in Table 1, except for the commercial ones [10,32]. Precipitation in KCl can be preferred to recover K2 with better gel characteristics (usually the precipitated carrageenan is less sulfated than the fraction remaining in the liquid phase [35,36,37,41,42]) or to partially get rid of agarans and other non-carrageenan products present in certain carrageenophytes. Alternatively, K2 can be simply recovered in film or powder form by water evaporation of the concentrated K2-rich solutions or by freeze drying [34,35,37]. The cooling of the concentrated hot K2-rich solution can also be performed to allow for gel formation. Various gel freezing/pressing/thawing cycles are then applied to expel water and other residues before drying. This method is, however, only amenable for K2 containing large K and salt contents to yield strong enough gels, and it was, indeed, not used in the studies referenced in Table 1. Depending on the employed separation and recovery protocols, different impurities are extracted together with K2. Thus, further K2 purification can be performed, for instance, by enzymatic treatment to get rid of Floridean starch [28,55]. In addition, K2′s chemical structure can be further modified (for instance, by alkali treatment to further reduce the number of sulfate groups on the polysaccharide chain [34,35,36,37,40,41,42]), a monocationic form of the polyelectrolyte can also be isolated (preferentially avoiding excessive dialysis to avoid a drop in the molecular mass [57]), or K2 can be further fractionated into less sulfated carrageenan by gelling into KCl.

Emerging greener extraction methods utilizing lower energy and water consumption, more ecofriendly solvents, and bio refinery concepts have also been tested with some of the seaweeds listed in Table 1. Microwave-assisted extraction [58,59] and ultrasound-assisted extraction [60] were shown to result in better K2 extraction yields, produce K2 with better gelling properties or, at least, to reduce the extraction time compared to K2 isolated with the conventional extraction routes provided in Figure 2. Extraction with subcritical water has also been tested but did not show promising results for K2 yields and gelling properties, at least for the seaweeds and the process parameters tested [61]. Biorefinery concepts were also applied to *M. stellatus* industrial wastes streamed from conventional extraction and showed that K2 and antioxidant compounds could be further extracted from the waste [62]. The scaling up and industrial intake of these greener processes are, however, still to be demonstrated for K2 extraction.

### 2.3. Statistical Block Copolymer Structure of Gelling Hybrid Carrageenans

The macromolecular structure of K2 has long been debated (see, e.g., [1]). Its labeling as hybrid carrageenan reflects the long discussion concerning whether K2 is a mixture of essentially K (homopolymer of ***G4S-DA***) and I (homopolymer of ***G4S-DA2S***) or a heteropolymer containing monomers such as ***G4S-DA*** and ***G4S-DA2S***. An historical perspective of this topic is perhaps the best way to discuss the polymer structure of K2, since the debate has been settled only a decade ago or so. 

In 1981, when Stancioff presented his diagram of K2 extracted from different seaweeds as a function of their sulfate content [63], he noted that K2 showed physical properties that differ from mixtures of K and I (K + I). Hybridity, in terms of the existence of heteropolymers, was recognized for K and I extracted from *Kappaphycus alvazerii* and *Eucheuma denticulatum*, respectively [64]; two seaweeds are mostly cultivated for the industrial production of K and I. Bellion et al. [64] found the biological precursors ***G4S-D2S*** and ***G4S-D2S,6S*** in carrageenans recovered after the enzymatic treatments of K and I, respectively [64]. Thus one could suspect a similar complexity in the macromolecular structures of K2 isolated from the seaweeds listed in Table 1. The nature of the hybridity was scrutinized by Rochas et al. [65] for the commercial carrageenan extracts from *C. crispus* and *M. stellatus*. Different macromolecular structures were hypothesized from mixtures of K and I homopolymers to copolymers showing a diblock structure or a statistical structure with short blocks of ***G4S-DA*** and ***G4S-DA2S*** randomly distributed along the polymer chain. The authors concluded from the ^13^C-NMR analysis of KCL soluble and insoluble fractions of carrageenans from *C. crispus* that both ***G4S-DA*** and ***G4S-DA2S*** were present in the fractionated products. However, these fractions were not submitted to the enzymatic treatment performed by Bellion et al. [64]. Thus, no conclusion regarding the copolymer nature of K2 from *C. crispus* and *M. stellatus* was delivered. 

A decade later, KCL fractionations under similar conditions were repeated by van de Velde et al. [66] for two commercial K2s. In addition, their solution properties were systematically compared with those of K and I mixtures (K + I) of corresponding composition in ***G4S-DA*** and ***G4S-DA2S***. KCl was not effective at separating fractions for both K2 in contrast to the corresponding K + I separated in a K-enriched gel phase and a I-enriched viscous phase. More important, the coil-to-helix transitions probed by intrinsic viscosity and optical rotation showed that K2 is characterized by a single broad transition in contrast to K + I, which showed the two transitions of both the I and K helices [65]. These marked differences prompted the authors to suspect that K2 was made of long blocks of ***G4S-DA*** and ***G4S-DA2S*** on the same polymer chain [66]. 

More recently, the statistical block copolymer nature of K2 (see Figure 3) was firmly established by W. Helbert’s group, who extended the enzymatic degradation studies of Bellion et al. [64] to a series of K2 and new ι-carrageenases and κ-carrageenases. 

**Figure 3 materials-16-05387-f003:**
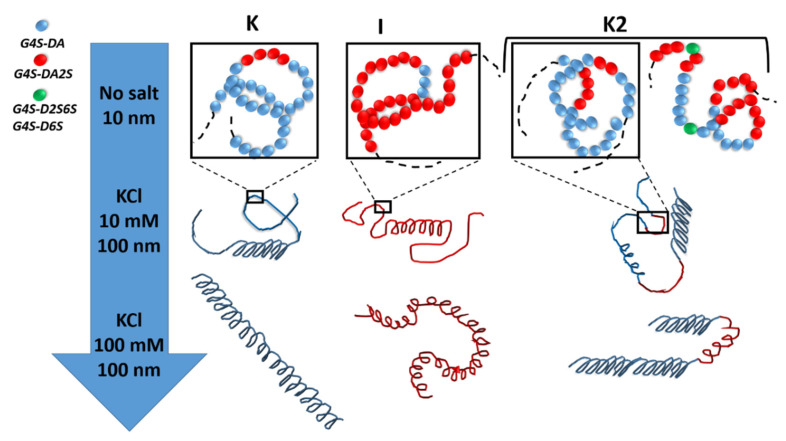
The statistical block copolymer structure of K, I, and K2. K and I from *K. alvazerii* and *E. denticulatum* contain 3 to 11 mol% of ***G4S-DA2S*** and ***G4S-DA***, respectively [10,60,62]. The sketches in the top row of the figure illustrates the block copolymer nature of K2 in the absence of salt (coil conformer), which includes sequences of ***G4S-DA2S*** (red sphere) and ***G4S-DA*** (blue sphere) interrupted by a more sulfated ***G4S-D2S,6S*** or ***G4S-D6S*** (green sphere), respectively. With the addition of salt, each block can adopt a helical conformation [54,55]. With more salt, full helices are formed with different levels of flexibility depending on the K or I block [67,68].

In two seminal papers reporting the in-depth chemical analysis of both degraded and undigested carrageenan products, Guibet et al. [32] and Jouanneau et al. [30] showed that K2 chains were made of (1) blocks of ***G4S-DA***, (2) blocks of ***G4S-DA2S***, and (3) sequences where ***G4S-DA*** and ***G4S-DA2S*** are randomly distributed. Moreover, the presence of biological precursor diads ***G4S-D2S*** and ***G4S-D2S,6S*** in blocks of ***G4S-DA*** and ***G4S-DA2S***, respectively, was demonstrated. These results, thus, confirmed earlier conclusions drawn from the modeling of coil-to-helix transitions in purified K2 (e.g., free of ***G4S-D2S*** and ***G4S-D2S,6S***) [55]. The analysis of the experimental data with a statistical block copolymer model suggested the existence of randomly distributed blocks of at least 8 to 14 ***G4S-DA*** diads and 2 to 5 ***G4S-DA2S*** diads, depending on the experimental technique used to probe the transitions and the type of salt used to induce the helical formation and aggregation. The lengths of the blocks are overall of the same order of the lengths actually inferred from the enzymatic degradation studies [30,32]. But more importantly, the latter underlined that the distribution of block lengths along the K2 chain varies from chain to chain and that this compositional variation is specific to the seaweed producing the K2, as well as the extraction process. In other words, and using a vocabulary specific to polymer science, K2 are statistical block copolymers with polydispersity in both chain and blocks lengths.

Several of the recent studies listed in Table 1 confirm the statistical block copolymer structure of K2 extracted under various conditions from different carrageenophytes. Essentially, the chemical analysis of the products from the acidic hydrolysis of K2 [29,37] or of fractionated K2 extracts with KCl [35,37,41] point towards a copolymer structure, as no K and I chains can be recovered separately. Furthermore, stepping on the Stancioff’s remarks [63], two studies highlighted the differences between the phase diagrams of several K2 and of the K + I with equivalent molar compositions in ***G4S-DA*** and ***G4S-DA2S* [69,70]**. On top of confirming that K2 was not a mixture of K and I, these results pointed towards the fact that phase separation between K2 and solvent occurs at ionic strengths and carrageenan concentrations that are specific to the K2 chemical structure and do not match those found for separating K and I from mixtures. This also explains why recent KCL fractionations of seaweed extracts in increasing ionic strengths are successful in separating K2 with different chemical compositions [35,37,41]. 

## 3. Gel Formation

To start with, it is convenient to review the current understanding of the gel mechanism in I and K. Upon cooling hot solutions of K or I in the presence of salt, a coil-to-helix conformational transition occurs. These helical secondary structures then self-assemble in networking superstructures that pervade in volume and provide elasticity to the network [1,10,11]. The exact nature of the helices and superstructures has long been debated, and several models have been proposed to relate the helical structure to the network elasticity (see, for instance, the following reviews and references therein [1,71,72,73]). The primary (coil), secondary (helices), and super-helical structures were only recently imaged by atomic force microscopy (AFM) of solutions and gels [67,68]. Single helices (i.e., intramolecular helices) are nucleated on the polysaccharide initial coil, as sketched in the second row of Figure 3. With the addition of more salt, more helices are formed on the macromolecule, resulting in a full helical conformation at larger ionic strength. Whereas rod-like helices were imaged for K, more curly ones were reported for I [67], as illustrated in the bottom row of Figure 3. Thus, both the charge screening effect, which is responsible for the improved flexibility in the coil state as more cation is added to the polyelectrolyte [74], and the specific nonmobile cation binding on the helical conformer [75] are responsible for the final helical conformation that will eventually self-assemble. Smectic-like domains (i.e., quaternary structures) of stacked super-stranded helices connected by helical branching to other domains were imaged in K gelled in the presence of potassium salt, whereas a mesh of intertwined single helices was imaged in I gels for all of the salts studied [68]. These morphological differences were used to explain the different levels of elastic shear moduli measured in the K gels and I gels. 

There are not many AFM images of K2 in the literature, but the few published AFM images and analyses suggest that K2 helices and assembly share similarities with those of K, with a little more branching and a less rod-like morphology [76]. As mentioned above, the occurrence of helical conformers of K and I in the K2 chains has been reported in differential scanning calorimetry (DSC) studies performed on heating K2 solutions in the presence of various salts [55]. As such, a sketch of K2 in the helical conformation is proposed in Figure 3. This sketch also builds on DSC and rheological data reported for a set of cooling K2 solutions in NaCl [54]. These K2 isolated from the alkali extraction from *M. stellatus* is in the Na^+^ form and contains between 31 and 38 mol.% of ***G4S-DA2S***, as well as a significant amount (up to 19 mol.%) of more sulfated biological precursors. A coil-to-helix transition was measured at a temperature *T_CH_*, which coincides with the first step increase in the shear loss modulus G”, measured in small amplitude oscillatory shear (SAOS) rotational rheometry. At intermediate ionic strength, namely, 0.1 M NaCl, a second step in the temperature dependence of G” is measured at temperatures below *T_CH_*. After gel maturation at 15 °C, the gelled samples show two helix-to-coil transitions in DSC curves measured upon heating, thus confirming earlier results [55]. However, the rheology showed that these transitions occurred at temperatures above the full melting of the gel. More importantly, the temperature dependence of G” showed two steps located before and after the gel melting temperature, suggesting the melting of two types of helical assemblies [54]. Examples of two-step gel setting rheological curves are given in Figure 4 for one of these K2 [54] in the presence of NaCl [77,78]. Cooling curves presenting two steps in both G’ (the shear storage modulus) and G” have also been reported for a different K2 at intermediate ionic strength in NaCl [79]. Indeed, both K2 concentration and NaCl ionic strength are essential to shift the two steps within the temperature window available in DSC or rotational rheometry. For instance, a rheological investigation of the cooling of a 1% solution of a K2 containing 50 mol.% ***G4S-DA2S*** in 0.2 M NaCl did not show two steps [80]. However, whereas the K2 in the Na^+^ form did not show more than a 1 °C difference between the gel setting, *T_g_*, and the gel melting, *T_m_*, temperatures, the K^+^ form exhibited a thermal hysteresis of 13 °C between the gelling and melting temperatures [80]. This result highlights the importance of the counter-ion in the gel formation of K2, which possibly stems from the cation specificity of K [1]. In the presence of KCl or CaCl_2_, a single step in the rheological functions is usually seen [33,78,81,82,83,84]. The single step in the steady shear viscosity during the cooling of a K2 in 0.05 KCl is illustrated in Figure 4c. The two-steps gelation of K2 seems, thus, specific to Na^+^ and points towards the independent coil-to-helix transitions of I and K blocks on the K2 chain, followed by the self-assembly of the helical structures.

This is reminiscent of the two steps reported for K + I in the presence of various salts and ionic strengths [85,86,87,88,89,90,91]. In these mixtures, each component exhibits separate coil-to-helix transitions that connect to the two-step increases in the viscoelastic moduli. Interestingly, the temperatures for the step increases in the moduli coincide with the temperatures measured in K or I solutions with corresponding concentrations [86,88,91], suggesting that the conformational transitions and subsequent helical aggregation are not perturbed by the other component in the blend. The same conclusion was reached for K2, as the helix-to-coil transition temperatures of the K and I blocks are the temperatures found in K + I of similar molar compositions [55]. The gel formation in K2 thus resembles the gel formation in mixtures of K + I with essentially the independent coil-to-helix transitions of each types of block and their self-assembly. However, these similarities do not help in identifying the nature of the self-assembly and the super helical structures, and even less in understanding the origin of the elasticity of K2 gels. This is because of the current debate on the structure of K + I gels. Microphase separation (at least at length scales above 100 nm [90]) between I-rich domains and K-rich domains has been suggested based on particle tracking [90], polymer diffusion in gels [89] and modeling of rheological data [86,87,88]. Alternatively, the co-aggregation of K and I helices forming super structures that are different from those formed in K gels has been suggested to explain the gel elasticity of K + I that is larger than the sum of individual gels elasticities [91].

Clearly, additional comparative studies on the cooling of K2 and K + I for different total carrageenan concentrations, salt types, and ionic strengths are needed for a better understanding of the formation of super helical structures and underlying elasticity. The following key features of K2 gel formation discussed in this section can however be highlighted here to inspire future studies. First, the gelling temperature, *T_g_*, commonly defined by the cross over between G′ and G″, is found to be independent of the amount of K in K2 when gelling occurs in the presence of KCl [55,81]. However, when NaCl is used, *T_g_* shows a greater sensitivity to the chemical composition of K2 and in particular to the molar content in ***G4S-D2S*** or its biological precursor [28,39,54]. Second, two-step gel setting is usually measured only when NaCl is used to promote the super helical assembly in K2 [54,77], whereas two-step gel setting is reported in many studies where K + I are gelled in KCl under ionic conditions where I helices self-assemble in a matrix of K coils [88,89,90]. Thus, NaCl seems a better choice to unravel the elasticity assigned to the self-assembly of I helices, since the larger elasticity of K super helical structures can rule the rheological properties of K2 or K + I. Third, since K and I gels show a hierarchical multilength scale structure, the rheological characterization of the gel formation in carrageenan should be accompanied by in situ structural characterization probing length scales, such as the size of the ***G4S-D2S*** or ***G4S-DA*** blocks, the size of the supercoiled helical assembly of I, the sizes of smectic-like domain of K, and then the microscale of their network. 

## 4. Linear Viscoelastic Properties of K2 Gels

Though K2 gels are known to show intermediate elastic properties between K and I gels [9,10,78], the literature reporting on their linear viscoelastic properties is far less extended than for K or I gels. Overall, the mechanical spectra of K2 measured by SAOS are qualitatively similar to the mechanical spectra of K or I. An example for K2 is provided in Figure 5a depicting a near frequency independent elastic modulus G′ and a loss modulus G″ showing a local minimum. The K2 sample was extracted from *M. stellatus* following a protocol detailed elsewhere and resulting in the recovery of a K2 in the K^+^ form with 68 mol.% of ***G4S-DA*** [39]. This K2 forms a turbid gel when cooling a hot solution of 1 wt.% of polysaccharide in 0.1 M KCl [69]. The K2 gel was formed in the Couette cell of a rotational rheometer (MCR300, Anton Paar) using the same cooling procedure as in [39]. Figure 5a also highlights the absence of any noticeable water synaeresis inherent to the volume change occurring at the liquid to gel transition [11,33]. This is in contrast to the important release of water from K gels, which adds complexity to their rheological characterization at large concentrations and ionic strength, in particular in KCl (see Figure 5b).

The water layer formed between the gel and the shearing surface promotes slip, which has a direct impact on the values of the measured moduli. Slip has a greater effect on sheared thinner samples [92]. This is illustrated in Figure 5b where two spectra of a K + I sample having the same ***G4S-DA*** composition as the K2 sample in Figure 5a are presented. The K + I sample was gelled under similar conditions but in two different shearing geometries. The measured spectra show shear storage moduli G′ differing by one order of magnitude, which is symptomatic of a more pronounced slip on the thinner gel formed in the Couette cell. The K2 gels did not show water release, and the gels formed in the rheometer readily stuck to the shearing geometry. This is shown in the picture in the inset to Figure 5a showing the shearing cylinder with stuck gel material after removal of the cylinder from the cup. Most of rheological studies documented the absence of synaeresis for K2 gels [28,39,48,54,59,69,70,79], unless they were ruptured [83]. But as for other carrageenan gels, K2s gel’s linear viscoelastic properties depended on various parameters. 

### 4.1. Impact of K2 Chemical Composition on the Gel Elastic Modulus

Since the first report by van de Velde et al. [55] showing a monotonic increase of G′ with the molar fraction of ***G4S-DA*** for a series of commercial K2 free of any biological precursors, very few studies reported on the gel viscoelastic properties of K2 as a function of the molar content in disaccharide units. In regard of the whole set of papers on the rheological properties of K2 reviewed here [28,33,39,48,54,55,69,70,77,78,79,80,81,82,83,84], it is clear that this topic has been overlooked. Moreover, a comparison of all of the referenced results as a function of the documented K2 chemical structure will be useless. This is because the gels were measured under different conditions (K2 concentration, ionic strength and type of salt, protocol for gel formation, etc.) which affect the gel linear viscoelastic properties (see sections below). We are, thus, left reviewing the few studies dedicated to the interplay between K2′s chemical structure and the gel elastic modulus.

In three studies where a series of K2 was extracted in the K^+^ form from *M. stellatus*, *C. crispus* and *A. devoniensis*, G′ of K2 gels in KCl was found to be negatively correlated with the molar content in ***G4S-D2S,6S*** [28,39]. A more recent study with natural K2 extracts (no use of alkali) from *S. skottsbergii* and *S. crispata* also positively correlated G′ measured in gels formed in KCl with the amount of ***G4S-DA*** in the K2 chain [83]. These results, thus, seem in harmony with the trend proposed earlier, naturally suggesting that stronger gels are formed when K2 is closer to the chemical composition of K [55]. In contrast to this, no correlation was found between the shear elastic modulus of gels in NaCl and the chemical structures of the K2 extracted in the Na^+^ form using NaOH or Na_2_CO_3_ [28,39,81]. Note, however, that in these studies, the range of variation in the content of ***G4S-DA*** was of the order of 15 mol.%, and that the K2 was extracted with different molecular masses *M_w_* [81]. The size of the K2 chain is known to impact on the elasticity of K2 gels formed in NaCl: longer chains lead to gels with lower G′ [48,54,81]. Thus, variation in *M_w_* could have screened out the effects of a small variation in the ***G4S-DA*** content. However, van de Velde et al. [55] also studied K2 with different *M_w_*, but no impact of the K2 molecular mass on the measured G’ of gels in KCl was discussed. This is possibly because the reported *M_w_* were all above the cut-off *M_wC_* beyond which K gel’s elasticity is constant in KCl [93]. The different trends documented here for the impact of K2 chemical composition on the gel elastic properties in NaCl and KCl call for a more specific review on the effect of salt on the viscoelastic properties of K2 gels.

### 4.2. Effect of Salt Type and Ionic Strength

It is known that certain cations are specific to K: stronger gels are formed in KCl or CaCl_2_ when compared to NaCl [1]. Note, however, that such cation specificity does not exist for I, especially when the carrageenan has been purified to get rid of any ***G4S-DA*** [85]. Nonetheless, due to the definition of K2, the type of salt and ionic strength are expected to play a major role in the gel characteristics since, as mentioned above, K2 with different chemical compositions can be recovered using successive KCl fractionations [35,37,41,65], and salt type has a strong impact on the gel formation (see Figure 4 above).

The effect of CaCl_2_, KCl, and NaCl on the elasticity of gels produced with a K2 containing 62 mol.% ***G4S-DA*** has been studied by Torres et al. [33]. The rheological characterization of the gels showed that the strongest gels were obtained with CaCl_2_, whereas the weakest gels were produced with NaCl. For each salt, a monotonic increase in G′ with ionic strength was reported. Moreover, the range of the K2 concentrations and ionic strength for which a gel is produced is very different for each salt. The phase diagrams produced for the three salts showed that K2 concentrations larger than 1.5 wt.% and at least 0.5 M NaCl were needed to produce gels, whereas a gel readily forms at 1 wt.% K2 in 0.1 M CaCl_2_. Similar phase diagrams in NaCl and KCl were reported for three K2 containing from 48 to 78 mol.% ***G4S-DA*** and extracted in the Na^+^ or K^+^ form [69,70] to avoid any possible synergistic effect of the cations on the gel elasticity [55]. The phase diagrams in KCl are reproduced in Figure 6. 

The figure clearly indicates that different gels, clear or turbid, are produced when using different ionic strengths and K2 concentrations. In particular, travelling along horizontal lines, that is increasing the ionic strength at a constant K2 concentration, different types of gels can be produced eventually leading to the formation of particulate suspensions for K2 having more than 50 mol.% ***G4S-DA***. Thus, after showing the monotonic increase in G′ with the ionic strength reported above for various salts [33,48,69,70], a maximum G′ is reached, followed by a drop. Similar effects were found for K gels, see, e.g., [94] for a recent account on the effects of cations and mixes of cations. But for K2, the location of the maximum in G’ shifts to larger ionic strengths when less ***G4S-DA*** is present in the K2 chain, see, e.g., the data for 1 wt.% in Figure 6. In other words, this indicates that K2 gels can retain more KCl salt than K gels, which is attractive for some applications. But, overall, the cation specificity reported for K2 stems from the cation-specific aggregation process of K helices [1].

### 4.3. Concentration Scaling of the Gel Elasticity

As recently noted in a review on the rheological properties of K and I gels, the gel elastic modulus increase with the polysaccharide concentration has been described by different laws and in particular by power laws with exponent values showing a large diversity [72]. In spite of the small amount of published concentration scaling in K2 gel elasticity, a similar diversity is found and even exponential dependence with the K2 concentration has been suggested [48]. Such diversity comes as no surprise given the variety of K2 composition in ***G4S-DA***, and the different salt conditions used to prepare the gels. Table 2 summarizes the values of the exponents reported in the short list of literature [69,70,72,78]. 

A general trend, however, is shown in Table 2: as the ionic strength increases, so does the exponent of the power law. In light of the phase diagrams shown in Figure 6, such a steeper dependence on the K2 concentration can be assigned to the formation of more turbid gels. Table 2 also shows that values for the power law exponent *n* are smaller when K2 is gelled with NaCl. Note, however, that such a conclusion does not hold for the listed K and I, possibly because of the mixture of salts which is inherent to these commercial products. 

Several models are available in the literature to extract some gel structural information from the exponent *n*. Taking on board the structural information reviewed in Section 3 and suggesting that, like K and I, K2 gels are made of interconnected stranded helices, one may focus on theories describing the elasticity of filamentous networks [95,96,97,98]. For these theories, *n* can vary between one and five depending on the rod-like or worm-like nature of the filaments as well as on the type of crosslinks fixing the filaments on the network. In light of these theories, an exponent *n* of the order of one found for K2 gelled in NaCl is associated to semiflexible filaments with finite extensibility that are prestressed by the crosslinks on a cubic lattice [95]. The exponent *n* of the order of 1.5, found at lower ionic strengths in KCl for K2 with more than 50 mol.% ***G4S-DA****,* corresponds to filaments of a rod-like character connected by links showing some flexibility [95]. Exponents of the order of two are computed for rod-like filaments showing both bending and stretching [97] or connected through rigid crosslinks [96], whereas exponents closer to three relate to more worm-like filaments networked by rigid crosslinks [96]. Note that no exponent *n* = 5 is listed in Table 2. Such a large exponent was computed for isotropic and concentrated suspensions of rods crowding a finite volume where physical contacts are responsible for the rise of solid-like elasticity [98]. Overall, the comparison of exponents *n* in Table 2 with the referenced theories suggest that the structure of K2 shifts from more flexible worm-like filaments at lower ionic strengths or in NaCl to stiffer rod-like filaments networked by rigid crosslinks. Thus, the concentration dependence of K2 gels’ elasticity connects with the structural information conveyed from AFM studies on K and I gels, as well as with the clarity or turbidity of gels reported in the phase diagrams of Figure 6: with more salt or K blocks, coarser structures are formed in the K2 gels, which are, thus, more turbid and show larger *n*.

## 5. Gel Properties under Large Deformation

In spite of the recognized importance of understanding the carrageenan gels mechanical response to large deformations [1,10], there are too few rheologically sound studies on the topic [68,72,82,99,100,101]. Rotational rheometry is, indeed, preferred to indentation testing (which is not reviewed here), since simple shear implies that symmetric tensors of deformation and stress are involved. This simplifies the computation of the rheological functions and then facilitates the theoretical interpretation of the results in the nonlinear regime of viscoelasticity, that is, when the shear modulus is no longer a constant. The gel’s rheological properties under large deformation are not only highly relevant for industrial applications, but they can also shed some light on the gel structure. This is because some of the theories reviewed above for rationalizing the concentration scaling of G′ by structural characteristics of the filamentous network also predict the gel’s elastic shear modulus behavior in the nonlinear regime of deformation [95,97,98]. Notably, all of these theories predict a strain hardening behavior for G′; that is, filamentous networks become stiffer under larger strain before rupturing [72]. The strain hardening originates from the finite extensibility of the worm-like filaments which are prestressed by the crosslinks of the network. Thus, both the bending rigidity of the filaments and the topology of the network modulate the stretch ability of the structure and, thus, the hardening [95,97]. In summary, the strand-like structure of carrageenan gels [1] should be reflected in the stiffening of the material under large deformation. 

The strain hardening of a K2 gel in KCl is displayed in Figure 7. The K2 sample is the same as the one tested in Figure 5a. Just after recording the mechanical spectrum displayed in Figure 5a, the gel was submitted to large amplitude oscillatory shear (LAOS) by sweeping the amplitude from SAOS to LAOS at a frequency of 1 Hz. 

**Figure 7 materials-16-05387-f007:**
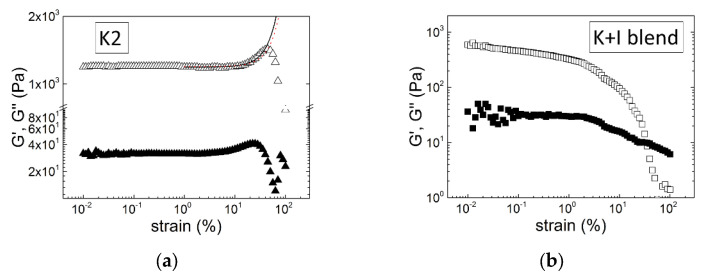
Large amplitude oscillatory shear tests (storage modulus G′, empty symbols; loss modulus G″, solid symbols, as a function of the applied strain) performed on gels formed in a Couette cell at 20 °C in 0.1 M KCl and with 1 wt.% carrageenan and characterized in Figure 5: (**a**) K2 extracted from *M. stellatus* and made of 68 mol.% of ***G4S-DA*** [69]; the solid line is computed from the exponential hardening model [102], whereas the red dotted line is computed from the filament-stretching-based model [95]; (**b**) blend of commercial K and I made of 70 wt.% of K.

Qualitatively similar strain-hardening curves were reported for I samples gelled in water in the presence of ethanol [98], for I gelled in NaCl [72], and for K2 samples containing a certain amount of ***G4S-D6S*** or ***G4S-D2S,6S*** and gelled in the presence of KCl [72,82,101]. Note that all of these gels showed rather soft elasticity characterized by a shear elastic modulus smaller than 1 kPa [72,82,99,101], as shown in Figure 7a, when compared to K gels in KCl, which can reach elastic moduli of the order of 10 kPa (see, e.g., Figure 5b). A softer gel elasticity is in harmony with the theoretical predictions for filamentous networks showing strain hardening, as it is needed to allow a transition from bending to stretching of the filaments under large deformation [95,97]. Figure 7a proposes a qualitative comparison between the experimental curve and the computations from an exponential form of the constitutive rheological equation [102] and from an equation derived from filament entropic elasticity and cubic lattice topological constraints [95]. The single parameter used to compute the exponential form is *l_max_*/*l*_0_ = 91, which suggests that the fully stretched length of the filament, *l_max_*, is much larger than its original length, *l*_0_, between two crosslinks in the network. This also indicates that the filaments built from the self-assembly of K2 helices show some flexibility allowing for significant stretching. The two parameters used to compute the filament-based model [95] are the filament stiffness parameter *e* = 0.15 and the end-to-end ratio *x* = 0.5, which relates the mesh size of the network to the contour length of the filament between two crosslinks. 

Among all of the nonlinear rheological studies listed above, three reported a strain softening for K [68,72,100], I [68,100], or even K + I [100] gels. A similar behavior is displayed in Figure 7b for the blend of K and I in KCl, which showed significant synaeresis (see Figure 5b). Thus, it is not clear whether the continuous drop in both G′ and G″ with increasing strain relates to the structure of the gels or is the signature of slip at the interface between the gel and the shearing surface. Clearly, as recently underlined [72], there is a need for additional experimental efforts in the rheological characterization of carrageenan gels under large deformation and for systematic comparative fits of available strain hardening theories to experimental data measured on a larger set of K2 gels. Such research will contribute to further explore the carrageenan gel structure underlying the elastic behavior at large deformation.

## 6. Conclusions and Perspectives

Although K2 are now clearly established as statistical block copolymers made essentially of sequences of ***G4S-DA*** and ***G4S-DA2S*** that are able to independently adopt helical conformations, there is still a long road to pave until a full understanding of the interplay between such macromolecular structures and the K2 gel’s properties. This review showed that K2 gels are different from gels made of blends of K and I with equivalent molar content in ***G4S-DA***. This was the starting point for deciphering the copolymer nature of K2, since free I chains are known to leak from K + I blends so that I can be separated from K under specific salt conditions in contrast to K2, which cannot be fractionated in K and I. However, reviewing the state-of-the art research on K + I, the actual debate on the microphase separated structure or co-assembly of helices in these carrageenan blends was also highlighted. 

This review also identified several controversial results on issues that still call for answers. These issues, detailed in the sections above, are wrapped up here. To what extent does the carrageenan composition of seaweeds match the chemistry of extracted K2? To what extent do the process parameters for seaweed grinding and drying affect the extraction yield of K2 and its chemical–physical properties? Is the gel setting temperature a function of the K2 chemical composition? Do K helices co-assemble with I helices, or are they separated assemblies of K and I helices? Is there a simple monotonic variation of the K2 gel elasticity with its content in ***G4S-DA***? If yes, can the variation be modeled? Are the gel properties independent on the K2 molecular mass *M_w_*, or is there a critical mass as for K gels? What are the K2 gel properties under large deformation, and how do they relate to gel structure and K2′s chemical composition? 

These questions call for future investigation of K2 and the following research strategies are proposed:-Focus on the study of K2 in the Na^+^ form and gel in the presence of NaCl. This is because the use of specific cations to K blocks smear-out the elasticity of I blocks in the copolymer. As a result, only the “weak kappa” character is unveiled, whereas in NaCl the elastic contribution of I blocks to the network are at least as strong as the K blocks contributions, resulting, for example, in the two-step gel setting;-Expand the study by van de Velde et al. [55] with more K2 with added complexity in the chemical composition. In the industry, many hybrid carrageenans are produced with significant levels of sulfated biological precursors and yet find application. Their texturing properties (as thickeners, not limited to gelling agents) also need to be understood based on their chemical attributes;-Focus on the effect of the molecular mass, *M_w_*, on the K2 gel properties. Since each seaweed produces copolymers with specific distributions of K and I blocks, it seems natural that different *M_w_* produced from the same parent K2 will have different effects on the gel properties unless *M_w_* reduction does not change the blocks distribution. So far, this topic has not been studied, but it should also contribute to the above point: establish the relationship between K2′s chemical composition and the gel elasticity without perturbation from *M_w_* effects;-Systematic comparative studies between K2 gels and K + I gels with similar ***G4S-DA*** composition and *M_w_* could be very instructive in understanding the block copolymer effect on the gel properties and, thus, contribute to settling the debate on K + I’s microstructure. This is because in K2, I cannot be separated from K;-Rheological and structural studies of carrageenan gels submitted to large deformation. As mentioned in Section 5, this topic has received nearly no attention, but it should bring an overwhelming amount of information on the elastically relevant structures in the gel, since theories are available to connect structures to the gel strain hardening. In addition, many applications involve the strong and fast deformation of gels and gel-forming solutions. There is, thus, an industrial need for such studies.

## Figures and Tables

**Figure 1 materials-16-05387-f001:**
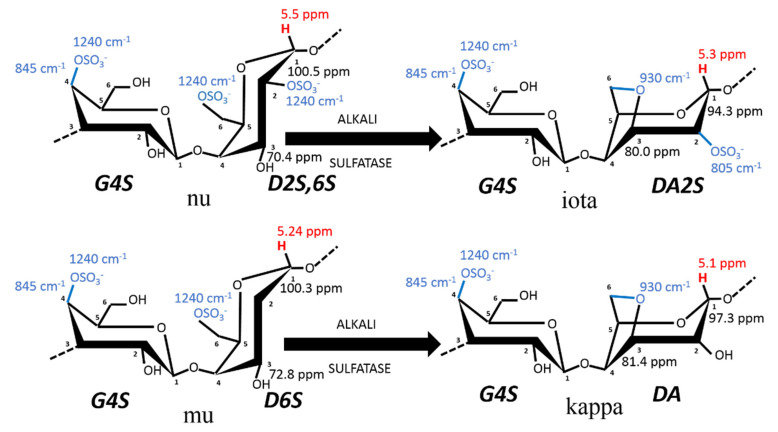
Chemical structure of carrageenan disaccharides comprising the sulfated polysaccharides K, I, and K2. K2 contains between 20 and 50 mol% of ***G4S-DA2S***, as arbitrarily set by the industrial definition of kappa-2 or weak kappa [9]. The diads given for each carrageenan structure follow the nomenclature introduced by Knudsen et al. [13], where numbers close to carbons indicate their numeration in the galactopyranose. For instance, in iota-carrageenan made of alternating residues of 3-linked β-D-galactopyranose (***G***) sulfated at the fourth carbon (***4S***) and 4-linked 3,6-anhydrogalactose-D-galactopyranose (***DA***) sulfated at the second carbon (***2S***) is noted as ***G4S-DA2S***, whereas in biological precursors, nu- and mu-carrageenan, the 4-linked α-D-galactopyranose is noted as ***D***, with the same nomenclature for indicating the positions of the sulfates. The transformation from (***D***) into (***DA***) can be performed in the industry by alkali treatment (ALKALI) of the seaweed or occurs enzymatically (SULFATASE) in vivo, to turn the biological precursors nu-carrageenan (N) and mu-carrageenan (M) disaccharides into less sulfated I and K, respectively. Note that ***G4S*** is common to all of these disaccharides and, thus, characterizes this family of gelling carrageenans. The blue numbers refer to the FTIR absorption bands, which characterize the corresponding chemical bonds highlighted in blue, whereas the red numbers correspond to the ^1^H-NMR chemical shifts of the corresponding anomeric proton highlighted in red. The numbers close to C1 and C3 of the 4-linked residues correspond to the ^13^C-NMR chemical shifts of the corresponding carbon.

**Figure 2 materials-16-05387-f002:**
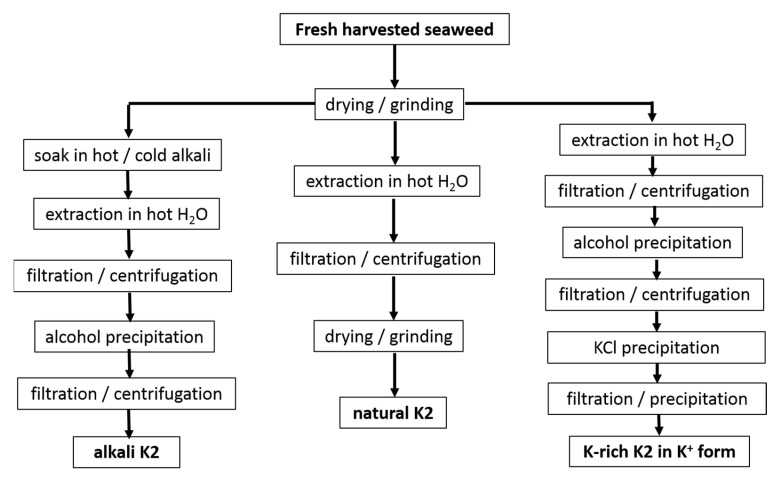
Protocols used in studies reported in Table 1 for the water extraction of K2 from seaweeds.

**Figure 4 materials-16-05387-f004:**
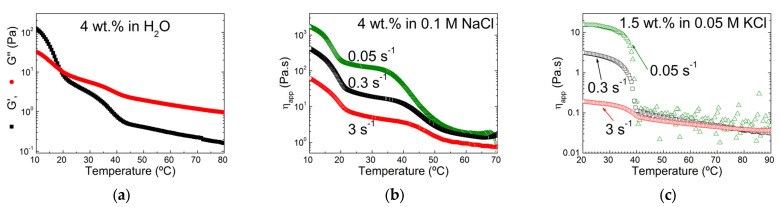
Cooling of hot solutions of K2 containing 38 mol.% of ***G4S-DA2S*** and 16 mol.% of ***G4SD2S,6S*** and ***G4S-D2S***: (**a**) temperature dependence of the shear storage (G′) and loss (G″) moduli recorded during the cooling of a 4 wt.% solution in distilled water under an oscillatory strain of 4%; (**b**) Temperature dependence of the apparent shear viscosity η_app_ measured during the cooling of a 4 wt.% solution in 0.1 M NaCl under a steady shear of 0.05 s^−1^ (green triangles), 0.3 s^−1^ (black squares), and 3 s^−1^ (red circles) [77]; (**c**) temperature dependence of the apparent shear viscosity η_app_ measured during the cooling of a 1.5 wt.% solution in 0.05 M KCl under a steady shear of 0.05 s^−1^ (green triangles), 0.3 s^−1^ (black squares), and 3 s^−1^ (red circles) [78].

**Figure 5 materials-16-05387-f005:**
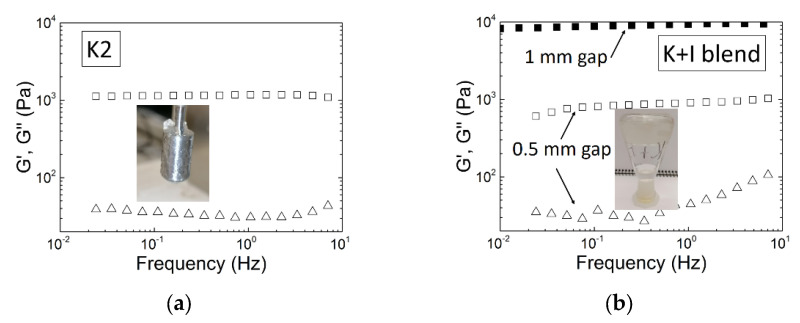
Mechanical spectra (storage modulus G′, squares; loss modulus G″, triangles) of gels formed at 20 °C in 0.1 M KCl with 1 wt.% carrageenan formed in the shearing geometry of a stress-controlled rotational rheometer (MCR300, Anton Paar): (**a**) K2 extracted from *M. stellatus* and made of 68 mol.% of ***G4S-DA*** [69], measured with a Couette cell imaged in the picture; (**b**) blend of commercial K and I made of 70 wt.% of K measured with parallel plates and a gap of 1 mm (solid squares) and with a Couette cell with a gap of 0.5 mm (open symbols); the picture shows the water synaeresis (bottom of inverted flask) from the gel phase (top of inverted flask).

**Figure 6 materials-16-05387-f006:**
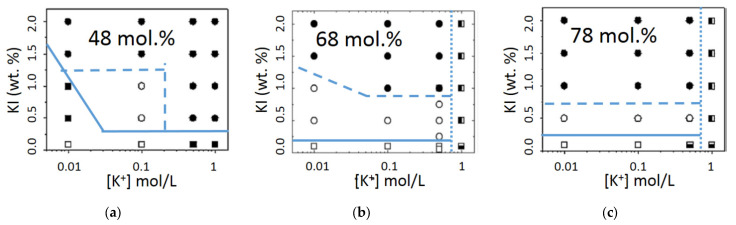
KCl phase diagrams (K2 concentration as a function of the ionic strength) [69]: (**a**) K2 with 42 mol.% ***G4S-DA2***; (**b**) K2 with 68 mol.% ***G4S-DA2S***; (**c**) K2 with 78 mol.% ***G4S-DA2S***. Solid lines signal the separation between solutions (squares, open for clear solutions, filled for turbid solutions) and clear gels (open circles), dashed lines separate clear gels from turbid gels (solid circles), and dotted lines separate the turbid gels from stable particles suspensions over 24 h (vertical semi-solid squares).

**Table 2 materials-16-05387-t002:** Exponents *n* of the power law equation G′ = *c^n^* describing the dependence of the linear shear elastic modulus, G′, on the carrageenan concentration, *c*, in gels formed under the conditions detailed in the column “Sample”.

Sample			*n*	Ref.
G4S-DA (mol.%)	Salt	Comment		
100	0.1 M NaCl	Commercial K—used as received	3.5 ± 0.4	[72]
100	0.02 M KCl	Commercial K—used as received	2.08 ± 0.13	[78]
78	0.01 M KCl 0.1 M KCl 0.5 M KCl 0.5 M NaCl 1 M NaCl	No mixed cations	1.42 ± 0.05 2.25 ± 0.07 2.98 ± 0.01 1.05 ± 0.01 1.23 ± 0.03	[70]
69	0.01 M KCl 0.1 M KCl 1 M NaCl	No mixed cations	1.37 ± 0.08 2.2 ± 0.1 1.18 ± 0.02	[69]
51.2	0.1 M KCl	With 31.7 mol.% of ***G4S-DA2S***	3.1 ± 0.1	[72]
51.2	0.05 M KCl	With 31.7 mol.% of ***G4S-DA2S***	3.18 ± 0.05	[78]
48	0.1 M KCl 0.5 M KCl 1 M KCl 1 M NaCl	No mixed cations	2.51 ± 0.04 2.14 ± 0.05 2.46 ± 0.08 1.01 ± 0.02	[70]
8	0.1 M NaCl	Commercial I—used as received	2.01 ± 0.08	[72]
8	0.05 M KCl	Commercial I—used as received	1.76 ± 0.05	[78]

## Data Availability

Data are available from the referenced studies or upon request for Figure 5 and Figure 7.

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
