# Peer review of "From Seaweeds to Hydrogels: Recent Progress in Kappa-2 Carrageenans"

_materials, 2023, doi:10.3390/ma16155387_

Round 1

Reviewer 1 Report

The manuscript summarizes the article "From seaweeds to hydrogels: recent progress in kappa-2 carrageenans." The scope of this review is specific and will be attractive to the Bio Polymers research community for understanding the rheological and structural studies of Hybrid carrageenans. The authors discussed the details of extracting hybrid carrageenans from seaweed, and structure analysis, including thermostability and rheological properties. However, the authors have not discussed any applications. This topic is very hot, and many other groups are working in this area. As a result, this review asks them to discuss the biodegradable polymeric applications of kappa-2 carrageenans. 

  1. What is the main question addressed by the research? The authors discussed extracting hybrid carrageenans from seaweed, and structure analysis, including thermostability and rheological properties.
2. Do you consider the topic original or relevant in the field? Does it
address a specific gap in the field? This report is relatively new, summarizing the details of the extractions of hydrogels from seaweeds, but more is needed for publication. It needs more information, including biopolymeric applications. 
3. What does it add to the subject area compared with other published
material? Yes, it is.
4. What specific improvements should the authors consider regarding the
methodology? What further controls should be considered? Carrageenans are important biopolymers and are widely used in various applications such as biomedical, environmental, agriculture, and food industry. I suggest authors to include some of them in this report during the revision process.
5. Are the conclusions consistent with the evidence and arguments presented
and do they address the main question posed? Yes
6. Are the references appropriate? Yes   7. Please include any additional comments on the tables and figures.

All these tables and figures are appropriate.

Author Response

COMMENT1: “ However, the authors have not discussed any applications. This topic is very hot, and many other groups are working in this area. As a result, this review asks them to discuss the biodegradable polymeric applications of kappa-2 carrageenans.” 

COMMENT2: “2. Do you consider the topic original or relevant in the field? Does it
address a specific gap in the field? This report is relatively new, summarizing the details of the extractions of hydrogels from seaweeds, but more is needed for publication. It needs more information, including biopolymeric applications.

COMMENT4: “4. What specific improvements should the authors consider regarding the
methodology? What further controls should be considered? Carrageenans are important biopolymers and are widely used in various applications such as biomedical, environmental, agriculture, and food industry. I suggest authors to include some of them in this report during the revision process.”

ANSWER: The reviewer is right in assessing that this review is very specific. Indeed, it focuses on algal sources, extraction routes, chemical structures, gel viscoelastic properties, and the interplay between all these topics, for a specific type of carrageenan. As such this paper is very different from the recent reviews on carrageenan which focus more on applications, and less on carrageenan itself, see e.g. the following reviews published in 2022 and 2023

-A Review of Carrageenan as a Polymer Electrolyte in Energy Resource Applications, http://dx.doi.org/10.1007/s10924-023-02903-0

-Recent advances in exploiting carrageenans as a versatile functional material for promising biomedical applications, http://dx.doi.org/10.1016/j.ijbiomac.2023.123787

-Biomedical and Environmental Applications of Carrageenan-Based Hydrogels: A Review, http://dx.doi.org/10.1007/s10924-022-02726-5

-Recent advances in carrageenan-based films for food packaging applications, http://dx.doi.org/10.3389/fnut.2022.1004588

-Carrageenan-Based Compounds as Wound Healing Materials, http://dx.doi.org/10.3390/ijms23169117

-Application of Carrageenan extract from red seaweed (Rhodophyta) in cosmetic products: A review, http://dx.doi.org/10.1016/j.jics.2022.100613

-Carrageenan From Kappaphycus alvarezii (Rhodophyta, Solieriaceae): Metabolism, Structure, Production, and Application, http://dx.doi.org/10.3389/fpls.2022.859635

-Carrageenans for tissue engineering and regenerative medicine applications: A review, http://dx.doi.org/10.1016/j.carbpol.2021.119045

We intend to maintain the specificity of our review, and thus step out from the recently published reviews mentioned above. From our understanding, a review on application would need a dedicated paper, with a systematic comparison between kappa-2 carrageenans and other carrageenans for each family of applications. Further, to cover topics such as biomedical, environmental and agriculture is beyond the field of expertise of the authors. Thus, reviewing kappa2 applications deserves a dedicated study with relevant experts in the field. We propose to leave this for another contribution, possibly within Materials with a dedicated issue.

Reviewer 2 Report

The authors presented a comprehensive review for the extraction of K2 carrageenans and the preparation of K2 hydrogels. Below are my specific comments:

1, In figure 1, please define what does G4S-DA2S refer to? Why is abbreviated in this term?

2, For table 1, for the seaweed species column, it looks too abundant. The replicates for the same species should be deleted.

3, For figure 2, there is no illustration for the above image. In addition, the quality of the image is poor.

4, In page 10, "viscosity during the cooling of a K2 in 0.05 KCl is illustrated in Figure 4(c)." The molar concentration unit is missing.

5, In page 14, "Table 4 also shows that n values are smaller..." it should be table 2 instead of table 4. In addition, what does n value refer to?

Author Response

COMMENT 1: ” In figure 1, please define what does G4S-DA2S refer to? Why is abbreviated in this term?”

ANSWER: We added the following lines in the figure’s caption to define the term: “….by Knudsen et al. [13]. For instance, in iota-carrageenan made of alternating residues of 3-linked b-D-galactopyranose (G) sulfated at the fourth carbon (4S) and 4-linked 3,6-anhydrogalactose-D-galactopyranose (DA) sulfated at the second carbon (2S) is noted as G4S-DA2S, whereas in biological precursors nu- and mu-carrageenan the 4-linked a-D-galactopyranose is noted D, with the same nomenclature for indicating the positions of the sulfates.”

COMMENT 2: “For table 1, for the seaweed species column, it looks too abundant. The replicates for the same species should be deleted”.

ANSWER: Redundant seaweed entries are now deleted, see new Table 1.

COMMENT 3: “, For figure 2, there is no illustration for the above image. In addition, the quality of the image is poor.”.

ANSWER: to avoid misleading information and poor quality of image, the latter is now replaced by a box with “fresh harvested seaweed”.

COMMENT 4: “In page 10, "viscosity during the cooling of a K2 in 0.05 KCl is illustrated in Figure 4(c)." The molar concentration unit is missing.”.

ANSWER: the molar concentration unit is now added and the line reads “The single step in the steady shear viscosity during the cooling of a K2 in 0.05 M KCl is illustrated in Figure 4(c).”.

COMMENT 5: “In page 14, "Table 4 also shows that n values are smaller..." it should be table 2 instead of table 4. In addition, what does n value refer to?”.

ANSWER: we corrected the line that now reads: “Table 2 also shows that values for the power law exponent n are smaller when K2 is gelled with NaCl”.

Reviewer 3 Report

1.    Why was the abscissa scale in Figure 4 (b) not linear?

2.    It was recommended to add the curve notes directly to Figure 4, 5 and 7.

3.    The text format and size in Figure 6 needed to be adjusted to be clearer and more appropriate.

4.    Please standardize the reference format.

1.    Please polish your grammar carefully. 

Author Response

COMMENT 1: “Why was the abscissa scale in Figure 4 (b) not linear?”

ANSWER: A logarithmic scale was used for the temperature for a better visualization of the 2 steps, in particular at lower temperatures. However, to address this comment, a new Figure 4 (b) is produced with a linear abscissa scale.

COMMENT 2: “ It was recommended to add the curve notes directly to Figure 4, 5 and 7.”

ANSWER: new figures are now produced with notes directly inserted in plots.

COMMENT 3: “The text format and size in Figure 6 needed to be adjusted to be clearer and more appropriate.”

ANSWER: all format and size for texts used in Figure 6 are now all consistent and larger, see the new produced figure which is clearer.

COMMENT 4: “Please standardize the reference format.”

ANSWER: the reference format is now standardized see corrections for references 10, 13, 15, 25,27, 51, 58-62, 64-71, 74-79, 81-102.  

COMMENT 5:  “Please polish your grammar carefully.”

ANSWER: thank you for the alert, the text has been thoroughly revised. 

Reviewer 4 Report

The review is a timely and detailled account of highly relevant information in the field of KA2 carragenans. The literature is celarly and critically discussed. 

Author Response

Many thanks for these positive comments.

Reviewer 5 Report

The name of most Fig and Tab in the manuscript are too long. The explanation of the Fig should be placed in the Fig or Tab, or a separate line of Note below the Fig or Tab.

Author Response

COMMENT 1: “The name of most Fig and Tab in the manuscript are too long. The explanation of the Fig should be placed in the Fig or Tab, or a separate line of Note below the Fig or Tab.”

ANSWER: following other reviewer’s comments, we inserted part of the information given in the captions to figures into the Figures 4, 5 and 7, but we also added information (see Figure 1). The text of Table 1 has been reduced to essential information which is now added into the table.